# The Role of Urban Cemeteries in Ecosystem Services and Habitat Protection

**DOI:** 10.3390/plants12061269

**Published:** 2023-03-10

**Authors:** Ágnes Sallay, Imola Gecséné Tar, Zsuzsanna Mikházi, Katalin Takács, Cecilia Furlan, Ulrike Krippner

**Affiliations:** 1Institute of Landscape Architecture, Urban Planning and Garden Art, Hungarian University of Agriculture and Life Sciences, 2100 Gödöllő, Hungary; 2Institut für Landschaftsarchitektur, Universität für Bodenkultur Wien, 1180 Vienna, Austria; cecilia.furlan@boku.ac.at (C.F.);

**Keywords:** cemetery, urban cemeteries, green space, urban green infrastructure, cemetery tourism, ecosystem services

## Abstract

Cemeteries, like urban public parks, are an important part of the urban ecosystem, providing semi-natural habitats for many plant and animal species as well as a wide range of ecosystem services: they improve air quality, reduce the urban heat island phenomenon and provide aesthetic and recreational value. This paper explores the role of the cemeteries in the green infrastructure network beyond their sacred and memorial role and their importance as a habitat for urban flora and fauna. In our study, we compared two large public cemeteries of Budapest (Nemzeti Sírkert/National Graveyard and Új Köztemető/New Public Cemetery) with the Zentralfriedhof Wien (Central Cemetery of Vienna), the latter of which has been forward-looking in terms of green infrastructure development and habitat creation in the past years. Our goal was to determine which maintenance technologies and green space development methods are most beneficial in terms of sustainable habitat creation and the use of appropriate plant species in public cemeteries.

## 1. Introduction

The expansion and densification of cities impose significant challenges for the urban environment: these are, amongst others, the loss of quality and quantity of green spaces, the decline of biodiversity, the degradation of ecosystems and the disconnection of human beings from nature, which affects their mental and physical health, including their well-being. When we talk about urban green spaces, we usually think of urban forests, tree alleys, urban parks and gardens, while cemeteries are frequently ignored. However, in the works of Sallay et al., 2022 [1]; Straka et al., 2022 [2]; Dlugoński et al., 2022 [3]; McClymont et al., 2021 [4]; and Nordh et al., 2018 [5], the topic of cemeteries as urban green infrastructure and recreational space has been unfolded (Figure 1). Going beyond, in this paper, we aim to analyze the role of urban cemeteries in biodiversity conservation and nature protection and the ecosystem services and habitat potentials they provide.

Cemeteries and burial grounds are primarily considered sacred places, closely linked to the history of a local community, but also custodians of natural and cultural values [6]. According to Skår (2018) and Maddrell (2016) [7,8], cemetery as a term refers mainly to the process of grieving and the dignified remembrance, but also to the management and maintenance of cemeteries [9]. However, cemeteries are an integral part of our neighborhoods, cities and society [10].

Over the past two centuries, land-use changes, rapid and accelerating urbanization, and population growth have led to a significant loss of natural habitats worldwide [11]. Due to this, the role of urban green spaces in providing semi-natural habitats has increased. In the changing metropolitan landscape burial grounds, cemeteries and churchyards might become islands of biodiversity conservation. Due to their size, habitat heterogeneity and continuity, cemeteries play an important role in urban biodiversity conservation [12]. Previous research has highlighted the important conservation functions and benefits of cemeteries in non-urban settings [4], which contribute to the preservation of natural habitats and rare species worldwide. Old cemeteries can provide “islands of habitat” for native species gradually squeezed out of intensively used rural landscapes, such as special endemic orchids in Turkey [13]. As stated by Löki and colleagues after reviewing 97 studies from five continents, cemeteries and churchyards have a significant conservation role. Even in highly modified environments, they often provide refuge for populations of rare and endangered species.

Cemeteries can provide important ecosystem services (ESs) such as support services (e.g., soil formation, photosynthesis, primary plant mass production, nutrient and water cycles), regulatory services (e.g., regulation of microclimate, of stormwater issues, of air and water quality) and cultural ecosystem services related to recreation, well-being and health [3,4,6,14]. ESs are the goods provided by living systems, and the human quality of life is determined by the sum of them. ESs can be classified into three groups [15]:Provisioning services: tangible goods provided by ecosystems that can be traded and consumed directly or indirectly, such as plant and animal food, raw materials and fibers, biomass and animal energy;Regulating and sustaining services: including how ecosystems regulate or modify biotic and abiotic environmental factors; they are not directly consumable or consumable goods, for example, bioremediation, flood control, erosion control, pollination, pest and disease control; soil formation and soil structure; climate regulation;Cultural service: intangible (non-tangible) assets that have symbolic, cultural or intellectual significance, such as tradition, heritage, aesthetics, recreation, leisure; educational–research activities [2]. Based on the ESTIMAP: Ecosystem services mapping at European scale methodology, urban green spaces contribute significantly to recreational opportunities [15].

A meta-analysis of 87 studies in 75 cities by Beninde and colleagues in 2015 revealed that the connectivity of habitats in cities is more important than the distance between green spaces. For contiguous habitats, an average minimum size of 50 hectares is required to conserve the rare species that live there [16]. The same study found that the level of damage caused by visitors, pets or road traffic and the level of disturbance caused by enclosure are lower than expected.

While there have been a number of studies on the contribution of urban parks to biodiversity conservation according to their size and land coverage [17,18,19,20], little is known about the similar role of cemeteries. Though urban parks and cemeteries share some common characteristics (e.g., existing woody and grassland habitat mosaics), there are significant differences in terms of the way they are used, the intensity of their maintenance and the specific “built elements” they include (e.g., high proportion of artificial surfaces in cemeteries due to paved surfaces and graveyard features). The role of cemeteries in relation to biodiversity is different from that of public parks, and this needs to be examined. The green spaces of cemeteries have a varying, but certainly measurable, impact on the local climate of their surroundings, mitigating the urban heat island effect. On a sunny, warm summer day, a cemetery can be several degrees cooler than its surroundings: the two largest cemeteries of Budapest (i.e., Nemzeti Sírkert and Új Köztemető) were respectively 1.7 °C and 7 °C colder than the neighboring urban areas, measured in August 2016, based on satellite and field surveys by Szent István University [21].

Cemeteries are dynamically changing landscapes; due to the continuous burials, their transformation might be more dynamic than that of public parks. However, cemeteries and especially their ecological or “green” conditions change gradually with each new burial. Their biodiversity largely depends on the character and size of the given cemeteries, their structural heterogeneity and, above all, the type and intensity of management [4,6]. If some areas of cemeteries are less intensively used or maintained, or even abandoned, this can lead to the development of a “novel”, more natural-like environment, allowing visitors to experience natural processes [2]. The management of cemeteries as living habitats has become increasingly popular in recent years; in the recent publication entitled “Der Friedhof lebt! Orte für Artenvielfalt, Naturschutz und Begegnung.” (The cemetery is alive! Places for Biodiversity, Nature Conservation and Social Exchange.), the author describes cemeteries as a paradise for flora and fauna [22]. The habitat characteristics of cemeteries are likely to be determined by the interaction between natural processes, management and design issues. The maintenance technologies and their intensity in the different areas of a cemetery have a strong influence on its ability to function as a habitat: the less intervention needed for maintenance, the more native plant and animal species can be found in the area [23].

This paper aims to fill the gap we have identified in this topic area by examining the potential of urban cemeteries for ecosystem services and environmental–biological values (Figure 2). We therefore address the following research questions:

What role does a cemetery play in urban green infrastructure?What habitat functions may cemeteries perform?Do cemeteries have specific ecological functions that differ from the potential of other urban green spaces?What ecosystem services do urban cemeteries provide? Do they differ in each case study?How do the three main functions of cemeteries (i.e., memorial, cultural/touristic and ecological) relate to each other? Do they reinforce or weaken each other? How can they be harmonized?How can the full potential of cemeteries be improved or exploited?

Many of the problems we have identified as research questions are not quantifiable and can only be measured at a normative level. Thus, we have chosen our research methods and materials accordingly.

## 2. Study Areas

Under the Austro-Hungarian Monarchy in the 19th century, the two cities of Vienna and Budapest experienced a similar development in the centralization of cemeteries. After the end of the Monarchy and especially after the Second World War, their development has taken different directions. By the 21st century, the use and development of central, large-scale cemeteries became significantly divergent.

### 2.1. Historical Context

A major change in the location of cemeteries in Hungary was brought about by the decree of Maria Theresa in 1775, in which the Empress prohibited burials around churches, which had been common practice since the Middle Ages, for public health reasons. Cemeteries were thus moved to the outskirts of the city. At the same time, the Hungarian capital was also undergoing enormous development, and Pest cemeteries moved further and further out over the next century. Yet the city caught up and overtook them: today, our largest cemeteries are integrated into its fabric, surrounded by houses. One of the oldest cemeteries in Budapest, still in operation today, is the Nemzeti Sírkert, a green space comparable in size to Városliget or Népliget, and only the Új Köztemető is larger. How these sites are viewed has also changed. Much larger areas than before were designated for the cemeteries, with plots, roads and junctions within them modeled on the urban spatial structure. Thus, the cemetery became a public pedestrian space, requiring a tree line and noble vegetation.

In the Middle Ages, Vienna, like Budapest, had several Christian and Jewish cemeteries. The cemeteries were subordinate to the parishes and were located around the parish churches within the city walls. From the 16th century to the 18th century, the monarch and the city administration sought to move the public cemeteries from the walled city to the suburban area, but this was only gradually achieved due to opposition from the ecclesiastical authorities. Spacious crypts (catacombs) were built under parish and monastery churches for the burials of wealthier people. The Josephine Toleration Decree (1781) meant that there was nothing to prevent Protestants from being buried in public cemeteries; in 1858, a separate Protestant cemetery was opened in front of the Matzleinsdorf line (Evangelischer Friedhof). These Josephite cemeteries were called municipal cemeteries after the takeover of the Vienna municipality in 1869. As the suburbs grew, space for the dead again became too limited. Hernals and Döbling, as well as the western suburbs, were given their cemeteries, of which the cemetery at Hietzing was particularly prestigious. The cholera epidemic of 1873 greatly intensified the debate on the relocation of cemeteries for health reasons. The five municipal cemeteries were no longer located on the outskirts of the town, as they had originally been, but in the center of the town. The most important innovation was the creation of the Zentralfriedhof, which was opened on 1 November 1874, next to the village of Simmering [24].

The Zentralfriedhof was not particularly popular at the beginning because of its distance from the city and its large size. To compensate for this, famous dead were moved there from other cemeteries, creating an attractive attraction with decorative plots.

In the Austro-Hungarian Monarchy, there was a rivalry with Vienna, Austria, German culture and the West on the part of the Nemzeti Sírkert. The Zentralfriedhof in Vienna was declared to be the nation’s cemetery, and its status was given immediately after its opening by secondary burials [25]. Although it was only opened in 1874, it developed much more dynamically, and by the 1930s it was much more orderly and impressive than the Nemzeti Sírkert. The generous design of its ornamental parcels is still almost unique in Europe.

### 2.2. Nemzeti Sírkert (National Graveyard), Budapest, Hungary (Figure 3)

Creation: The cemetery was opened in 1847 as a public cemetery in Pest, and by the end of the 19th century it had become the most prestigious memorial site in Hungary. When it was opened, the cemetery had a simple rectangular layout, and by the 1870s the main roads had been planted with trees, earlier than its grounds were designed and executed in its present architectural layout at the beginning of the 20th century. In 1885, it was already registered as an ornamental graveyard, before that as a public cemetery, so it was not originally founded as a memorial site of the nation. In 1952, the cemetery was closed and the liquidation of the cemetery began, and in 1956, the Metropolitan Council declared the cemetery a National Pantheon. In the 1970s, the cemetery was landscaped, with many graves being removed and replaced by landscaped green areas. In December 2013, the entire cemetery was declared a national monument and listed as a historical monument. In 2016, the National Heritage Institute was appointed as the cemetery’s administrator and trustee. It is one of the capital’s most important attractions, its splendor and richness reminiscent of the cemeteries of the world’s cities, both a beautiful park and a major pilgrimage site [25,26,27].

Environment, location and green space connections: The cemetery garden, known for its rich flora and fauna, covers 56 hectares, one of the largest contiguous green spaces in the capital. It is one of the largest cemeteries in Europe, with a large number of natural and architectural features, and is as important as green space as it is as a cultural space. As a green space, it is an important link in the “green arch” between the Városliget and the Népliget [25]. The condition of the park is also important for the city, and special attention is paid to the woody plants living there, including several veteran trees over a hundred years old [26]. The Jewish cemetery on Salgótarjáni Road, which was opened in 1874 and covered 4.8 hectares, is directly connected to the cemetery area and has a significant plant population and mature tree trunks.

Current state, flora and fauna: The National Heritage Institute considers the cemetery’s trees, planted in the 19th century, to be of outstanding historical value. The quadrangular and diagonal paths are flanked by rows of old sycamore (*Platanus* × *hispanica*), chestnut (*Aesculus hippocastanum*), hackberry (*Celtis occidentalis*), linden (*Tilia* sp.), acacia (*Robinia pseudoacacia*) and cigar tree (*Catalpa bignonioides*). The total number of plant groups in the tree-lined parcels exceeds 3500. Among the trees of unique value are the oak trees (*Quercus robur*) next to the tomb of János Arany. The back plots of the cemetery, which are richly vegetated, provide important habitat for fauna. The Hungarian Society for Ornithology and Nature Conservation has observed more than 100 species of birds living and 40 species breeding there (e.g., thrushes, barn owls, woodpeckers, wood sparrows, sparrows, wood owls). The flowers attract insects and insectivorous bats in large numbers. There are numerous species of small rodents in the undergrowth of the cemetery (including various species of mice, some of which are rarer, and also pelicans, the eastern hedgehog and woodland or red squirrels in the trees). Due to their increasingly scarce natural habitat, wild animals have recently been moving into the inner areas of towns and cities, and several fox families have moved into the cemetery. Red foxes hunt mainly at night, their main prey being mice and voles, but they also threaten ground-nesting birds [25,26,28].

Planning/development context: In 2021, the National Heritage Institute commissioned the preparation of a management plan for the Nemzeti Sírkert. Following a general green space assessment of the cemetery (vegetation, tree lines, spatial structure, road network, small architectures), the authors have formulated maintenance and development proposals. The ecological development strategy also aims to increase biodiversity and preserve and expand habitats:Low-impact conservation technologies, limiting the use of pesticides;Extensive “wildflower” grassland management, creation of flower meadows (planned expansion of extensively maintained grassland areas, which, in addition to increasing biodiversity, also serves to reduce the costs of operating green spaces);Management of bird and insect species;Ecological use of plant waste and straw, with on-site composting (shredding, mulching);In terms of flowering areas (in priority, super-intensive areas), the role and surface area of more ecologically and economically maintained perennial beds should be increased in preference to annuals;Awareness raising and information are a priority when introducing ecological green space management models [29].

**Figure 3 plants-12-01269-f003:**
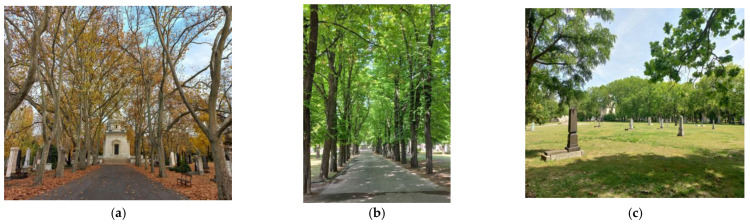
Nemzeti Sírkert, Budapest: (**a**) the sycamore alley leading to the Deák Mausoleum, (**b**) chestnut alley, (**c**) landscaped plot (authors’ photos, 2022).

### 2.3. Új Köztemető (New Public Cemetery), Budapest, Hungary (Figure 4)

Foundation: The cemetery, which has grown to 207,000 square meters, is one of the largest cemeteries in Europe. A significant part of it was designated as a military cemetery: this is where the present Heroes’ Cemetery was created [25]. The regular, geometric layout and the rectangular plot system formed by the rectangular road network were based on plans by the architect Győző Czigler in 1903, but some of the plots still have no graves. To date, some 3 million people have found a final resting place here. The victims of the conceptual trials of the 1950s and the 1956 revolution were buried in plots 298, 300 and 301 of the cemetery. In recent decades, this part of the cemetery has been tidied up, a visitor center has been built and it has been declared a national memorial site [30].

Environment, location + green space connections: The cemetery is located on the outskirts of Budapest, on the border of Kőbánya and Rákoskeresztúr. The Jewish cemetery of Kozma Street and the Orthodox Jewish cemetery of Gránátos Street are located next to the cemetery and also have significant vegetation cover. The Új Köztemető is bordered to the east by the Keresztúr Forest, which, in addition to its importance as green space, also serves the recreational needs of the residents of the area.

Current state, flora and fauna: Only about a quarter of the green areas in the vast cemetery are regularly maintained, leaving a large number of unmaintained, overgrown plots. Forest animals feel at home in the cemetery: there is a deer population of 50, and larger mammals such as foxes and hares are often seen. The cemetery is an undisturbed habitat for many species of birds (e.g., owls, ravens) and bats. In 2019, a nature trail, a jogging track and a cross-country skiing track were created in the Crossroads Forest next to the cemetery.

Planning/development context: no information available at present.

**Figure 4 plants-12-01269-f004:**
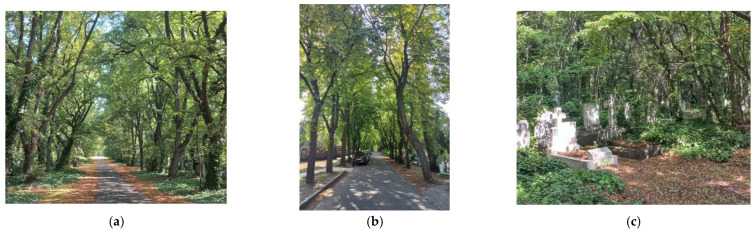
Új Köztemető, Budapest: (**a**) lime tree alley along one of the main roads in the cemetery, (**b**) chestnut tree alley, (**c**) wooded plot (authors’ photos, 2022).

### 2.4. Zentralfriedhof (Central Cemetery) of Vienna, Austria (Figure 5)

The Zentralfriedhof was established in 1870 by two architects, Karl Jonas Mylius and Alfred Friedrich Bluntschli. The cemetery was inaugurated in 1874. Mylius and Bluntschli’s neo-baroque design is based on the shape of a Greek cross. Each of the three arms of the cross culminates in a circular square from which eight streets are radiating, similar to a baroque etoile square. The church was projected at the intersection of the main axes but was only constructed in 1908–1910 following the design of Max Hegele. The cemetery was extended seven times until 1921. Today it covers an area of about 2.5 km^2^, making it the second-largest cemetery in Europe after Hamburg Ohlsdorf. It contains 33,000 gravestones. Over the years, some 3 million people have been buried here. Some 1000 of them are famous personalities, including all the Austrian presidents since 1945, buried in the presidential crypt in front of the Art Nouveau church. There are also sections for Catholic and Lutheran cemeteries, old and new Jewish cemeteries and Russian Orthodox cemeteries [31]. Most of the Zentralfriedhof is currently managed by Friedhöfe Wien GmbH. The Protestant and Jewish cemetery sections are exceptions. The Jewish cemetery is run by the Jewish Community of Vienna IKG, and the Protestant cemetery is run by the Protestant Church. Since April 2021, the University of Vienna, in cooperation with Friedhöfe Wien GmbH, has been conducting research under the project “BaF—Biodiversity in cemeteries” under the direction of project leader Thomas Filek. The aim is to study biodiversity and anthropogenic impacts on cemeteries as habitats [32].

Environment, location + green space connections: Contrary to its name, the Zentralfriedhof is located on the southeastern outskirts of the city, in Simmering, which was annexed to Vienna in 1892 and was not part of the city when the cemetery was built. The Zentralfriedhof is part of the eastern part of Vienna’s Green Belt, which was established in 1905. The Vienna Green Belt encompasses the city’s entire area and covers a total area of about 21,500 hectares. The Green Belt’s eastern part includes the Zentralfriedhof and the Prater and the Simmeringer Haide. Opposite the cemetery, near the Feuerhalle Simmering, is a historic oak forest, a natural monument already mentioned in a document from 1649 [33].

Current state, flora and fauna: The cemetery contains nearly 15,000 trees, a significant proportion of which is made up of alleys, trees along paths and the dense tree population in the area of the old Jewish cemetery. In 2009 and 2016, plots were transformed into forest cemeteries. The so-called Naturgarten was created in 2011 next to Gate 9, with a large elder tree (*Sambucus nigra*) as its centerpiece. This area covers 40,000 square meters. The area also includes a biotope and two butterfly fields to increase biodiversity. In the large open green areas where taller grass grows, signs indicate this is a “natural meadow for insects and their mates”. Due to its size, its location on the outskirts of the city and its diversity, the Vienna Zentralfriedhof is also of special importance as a refuge and habitat for many animal species, such as field cricket (*Gryllus campestris*), Viennese night peacock (*Saturnia pyri*), blue butterfly (*Polyommatus icarus*), little eider (*Coenonympha pamphilus*), green toad (*Bufo viridis*), sand lizard (*Lacerta agilis*), middle-spotted woodpecker (*Picoides medius*), field hamster (*Cricetus cricetus*), various species of bats [34], storks, badgers and martens. There are also many squirrels, which are often fed by visitors to the cemetery, despite the many signs warning visitors not to feed the animals, as inappropriate feeding can endanger their lives. The cemetery’s largest “animal residents” are the 20 or so roe deer, a favorite in the old Jewish cemetery, not least because of the evergreen plants growing around the old gravestones, which provide a reliable source of food, especially in the colder seasons. Field observations show that the biodiversity differs between parts of the cemetery; e.g., the old Jewish cemetery has become overgrown and is more species-rich, while the new Jewish cemetery, founded in 1911, is more regulated and in its character comparable with the non-religious part of the Zentralfriedhof [35]. Until the mid-1980s, the rich population of roe deer and other wildlife was officially controlled by hunting and a hunter appointed by the Forestry Directorate. Today, the City of Vienna’s environmental department, Network Nature, with its species and habitat conservation program, ensures that, in addition to the well-maintained walkways and graves, the overgrown, semi-natural areas are preserved [32,36] (p. 87).

Planning/development context: The management of the Vienna Zentralfriedhof has not published any strategic plans yet. However, the Urban Development Plan of the City of Vienna STEP 2025 highlights the significant impact of the Zentralfriedhof in classifying it as “large urban green space” of regional importance which serves as a fresh air corridor and shall be linked to neighboring green spaces [37].

**Figure 5 plants-12-01269-f005:**
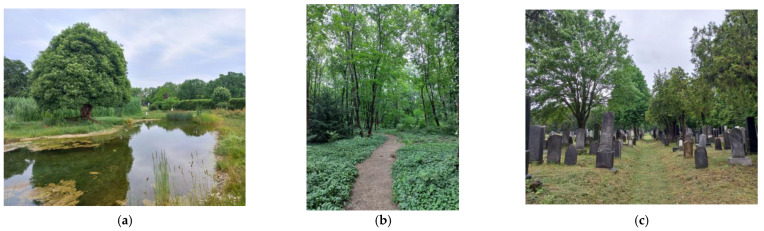
Zentralfriedhof Wien: (**a**) biotope in the Naturgarten, (**b**) path in the forest cemetery, (**c**) the old Jewish cemetery (authors’ photos, 2022).

## 3. Materials and Methods

Our study is the first step of long-term research, and our aim is now to gain a deeper understanding of the problems, processes and interrelationships. Therefore, the research focuses on a qualitative understanding of the baseline situation and on collecting and analyzing opinions and attitudes. In our studies, we have carried out a detailed exploration of the research area on a small sample; our aim is not yet to be representative. Interactivity and an inductive approach were the main features of this phase of the study, which was carried out using a qualitative technique, and one of the main tasks was to highlight individual characteristics. Typical qualitative research methods were used, such as observation, expert interviews and case studies.

To address the aforementioned questions, we reviewed the historical development and changes of the study areas of Budapest and Vienna, mainly in terms of spatial extent, subdivision and vegetation cover. A literature review showed that the role of urban cemeteries in biodiversity conservation is an under-explored and under-researched topic. For rural cemeteries, there is a much larger national and international resource base (Appendix A). Secondary sources in the literature were supplemented by field visits and personal visits to the cemeteries in all three sample areas.

### 3.1. Literature Review

The theoretical basis was determined by reviewing literature sources on the following topics: ethnobotanical research in Hungary; botanical and habitat research in cemeteries, by reviewing international sources; green space development concepts and studies related to cemeteries in Budapest.

#### 3.1.1. International Literature

In addition to reviewing the literature on the environmental values [4,6,38] and conservation functions [4] of cemeteries, considerable attention has been paid to sources that process and describe the special fauna and especially the flora of cemeteries [13,39,40,41]. In Austria, the Österreichischen Gesellschaft für Entomofaunistik (Austrian Society for Entomofaunistics) has several documents dealing with the assessment of the insect fauna of cemeteries [33].

We also addressed the use of cemeteries and newer burial methods with higher ecological benefits [42]. The works of Kowarik et al. (2016) [6] and Dlugoński et al. (2022) [3] served as a starting point for research on ecosystem services of cemeteries, while for urban habitats, we took the work presented in [43]. The role of urban parks in biodiversity conservation was based on the work presented in [19,20].

#### 3.1.2. Budapest Cemetery Development Concepts

Chapter II.7 of the Green Space Management of the Environmental Assessment of Budapest 2021 [44] shows the distribution of burials in Budapest, which is increasingly shifting towards cremation: 86% of burials are cremations (burial, scattering, urns) and 14% are traditional funeral services. The number of alternative burial services, such as biodegradable urns or the natural disposal of ashes (e.g., boat scattering, air scattering), has also increased significantly in recent years. There is also a growing demand for eco-friendly burial sites, such as forestry burials and memorial forests. In light of changing burial patterns, the document argues that it would be advisable to provide as many alternative burial methods as possible in the capital’s public cemeteries, involving the use of existing cemetery areas while at the same time meeting specific needs. One possible form of this could be the introduction of “forestry or natural burial”, with site areas that are currently part of the service. Plots with more than 65% of expired graves could be suitable. By clearing the wooded plots, areas that currently appear neglected could be brought back into service or, following their park-like restoration, the recreational role of the green space could be enhanced as a kind of memorial garden. In the case of cemeteries with expansion areas, it is intended to provide for the possibility of burial with wood instead of grave markers when new plots are opened. The document also envisages the creation of memorial forests in the near future, using existing woodland within the capital. Both wood burials and forest cemeteries and memorial forests provide a high proportion of green space, which can increase cemeteries’ biodiversity and habitat function. The Metropolitan Cemetery Development Study, which incorporates the above principles, was completed in 2019 and is expected to be adopted and implemented in the near future.

The Action Area X of the Radó Dezső Plan [45], approved in 2021, concerns the cemeteries of Budapest, with the following objectives: The main tasks related to the achievement of the objectives, which are also relevant to the present research, are “Better use of the recreational potential of existing cemeteries; public use of closed cemeteries; temporary green infrastructure use of disused cemeteries” and “Promotion of alternative burial methods; designation of urban areas suitable for forest burial” [45] (p. 62). The Capital intends to complete the planned improvements to the cemeteries by 2027.

#### 3.1.3. Vienna Cemetery Development Concepts

The strategic concept on the development of Vienna’s green infrastructure system, published in 2014 by the Vienna City Administration, aims at increasing the livability of the city by protecting and improving the connection between existing elements and by creating new green infrastructure units. The distance between each green space element shall be reduced to 250 m. The concept classifies the Zentralfriedhof as a semi-public green space responsible for the following:Everyday recreation;Structurizing the urban fabric;Providing ecological services;Providing pedestrian and cyclist passage;Potentially creating habitat [37].

### 3.2. Interviewing Cemetery Managers

There are many efforts or activities that are not visible or visible to the visitor and have not been published but are essential for habitat function and to provide constant ecosystem services to the surrounding urban structures. These hidden activities have been unfolded through a series of structured interviews with cemetery managers on the subject.

For all three cemeteries, the same questions were asked in writing to the cemetery managers and maintainers. The method was open-ended, with the interviewees answering the questions freely, in their own words, and at any length. Because of the written nature of the interviews, we employed two types of questions: short, factual questions and longer, more elaborative questions:How compatible do you see the memorial function of cemeteries with the development of the cemetery as a habitat?Are there, or have there been, any surveys of the cemetery’s wildlife (plants, animals)?Is the cemetery’s development taking into account, and seeking to create, habitats? If so, which developments are concerned with this?When maintaining the green areas of the cemetery, is it a consideration or an effort made to create habitats? If so, how and what has been changed?Is there a follow-up on the impact of the developments on wildlife? Have they looked at how the composition of flora and fauna has changed?Are there any developments that did not work or did not have the intended effect?How has the development been received by the public? Has the group of visitors to the cemeteries changed due to the developments? If so, how? Who, if any, newcomers have come and who, if any, have left?Do you have relationships with partners who prioritize protecting and enhancing the habitats and wildlife (plants, animals)?

## 4. Results

We examined the conditions of the Nemzeti Sírkert and the Új Köztemető regarding habitat design and habitat protection. We compared them with the solutions found in the Vienna Zentralfriedhof. The results of the literature research, the interviews and the site visits were summarized in a table, which clearly shows the advantages and shortcomings of each cemetery.

### 4.1. Results of Interviews

Nemzeti Sírkert (managed by NÖRI (public body of the government), data provider: János Prutkay, Deputy Service Director): The memorial function is only one of the main features, but also the educational, touristic, cultural and sacral functions are considered important, for which an appropriate and attractive natural environment is sought. The cemetery function in the traditional sense is only present in one part of the cemetery. Some areas are now less frequented, and, consequently, the natural habitat character has been strengthened. In terms of plants, an inventory of woody plants is in place and is being updated. The Hungarian Ornithological Society carried out a survey of the cemetery’s bird population, according to which 110 species were observed in the cemetery. The Hungarian Ornithological Society has installed 25 B-type bird boxes and a feeder and monitors the bird species present in the cemetery, occasionally carrying out ringing. Feeding of the birds is carried out by the maintainer. A distinction is made between intensively and extensively maintained areas. Extensively maintained plots are disturbed only to the extent necessary, with only roadside mowing and no disturbance to habitats within the plots. There have been no recent developments of a comprehensive nature in the cemetery that would have had a significant impact on habitat design or wildlife. In NÖRI’s experience, the green space improvements have been welcomed by the public, and feedback indicates that they are satisfied. Thanks to the improvements, and largely thanks to the NÖRI’s Remembrance Education Programme, the visitor groups have increased, including schools and groups of Hungarians from outside the country. There is a good relationship with the Rákóczi Association, which brings many school groups to the cemetery.

Új Köztemető (maintained by BTI (public body of the metropolitan municipality), data provider: Ádám Horváth, Head of Division): In the experience of the cemetery’s management, visitors to the cemetery would find it difficult to accept improvements to the habitat, and so there are currently no such developments in the Új Köztemető. They do not rule out the possibility of such improvements in the future, after careful consideration, in certain parts of the cemetery. They do not have any data on the cemetery’s livestock, but a student is currently studying the deer on the cemetery’s grounds as part of a thesis supported by the City of Budapest. A cadastral survey of the trees in the cemetery, especially the row trees, is available. Although there is no record of the whereabouts of the birds’ nesting sites, i.e., the cemetery tree survey (tree register) does not cover nesting sites, efforts are being made to protect them. During the molting season, only urgent tree care work is carried out, which is also agreed upon with the tree care professionals working in the area.

Zentralfriedhof (managed by Friedhöfe Wien GmbH, data are based on an interview by [32]): Political and economic factors play a major role in managing the cemetery, but biological aspects are also taken into account. The cemetery management is aware that it has a big task ahead of it in terms of sustainability and biodiversity protection in one of the largest green areas of Vienna. In 2000, a tree register was established in the Zentralfriedhof, recording the trees’ age, species and condition. Of the trees in the cemetery, 1200 are over 100 years old, according to the register. Several surveys of the cemetery’s fauna have been carried out, with around 100 vertebrate species, a third of which are birds. A census of insects, mollusks and herbaceous plant species is planned for the coming years. Every year, many graves are removed from the cemetery and efforts are made to systematically cover the freed areas, taking care to create contiguous areas. If, for example, two rows of graves become completely empty, resting places are created there. If whole groups of graves are cleared, larger areas can be freed up. There are many green areas where there are currently only a few rows of graves along the roads, which were previously covered by graves. There is scope to create alternative grave types. New trees will be planted not far away during the replacement planting to create an extension area for the Waldfriedhof. They try to make as little intervention as possible, letting the forest grow naturally.

The cemetery has 80 km of roads; there is a lot of pavement, and the three etoile squares and the square in front of the church count for a lot of paved area. Access by car is allowed on paved routes. Maintenance involves mowing the paths at least nine times a year, considering vegetation growth. The main concern is accessibility and safety. Unmaintained graves are mowed at least three times a year, where aesthetics are the deciding factor. Between graves, mowing is carried out together with the graves. The meadows along the paths are mown nine times a year, as are the paths. Large open meadows not directly used are mown twice a year. They are also marked with signs as natural meadows. These green areas are deliberately left as habitats for various insects. The mowing schedule is based on recommendations from experts.

The trees are usually inspected once a year; if an anomaly is found, it must be rectified within a reasonable time. The watering of the plants can be a problem as the cemetery faces shortages of rainwater due to climate change. The officers will set a date for the next pruning and check that the work has been done. The health of the trees is also important for safety reasons so broken branches do not cause accidents during major windstorms. The cemetery is participating in a program to preserve dead trees. The cemetery has committed not to remove all dead trees but to retain them where possible as valuable habitats for insects.

Environmental protection has become increasingly important in cemeteries in the last few years. Visitors are reminded that much of the water provided in cemeteries is drinking water, which can contribute to more prudent use. Waste is collected separately. In addition, battery-powered devices are increasingly used in the cemetery. E-vehicles have also been used for years as small means of transport. Using battery-powered devices has also reduced noise levels in the cemetery, which is beneficial for wildlife.

To promote the conservation of the species, a hedgehog conservation project has been launched: leaf piles are left in the autumn to provide a habitat for hedgehogs. Nesting boxes have also been placed for the birds. In the cemetery, a large green area has been set aside as a “Netzwerk Natur” area. “Netzwerk Natur” is the Vienna Species and Habitat Conservation Programme, which contains standards and guidelines.

The Park of Peace and Strength (Park der Ruhe und Kraft) was the first development designed to enable visitors to interpret the cemetery as a place of encounter. Several programs are linked to the development, including the possibility of remembering artists on the graves of honor, attending a cemetery concert or visiting the funeral museum, and meditating in the park. Vienna’s Zentralfriedhof is very popular as a local resting place for the city’s residents. Many visitors come here not to mourn, visit or tend graves, but to relax, unwind and find peace. A cemetery is a popular place for hiking and photography. The cemetery is generally used intensively, and the dilemma of memorial and recreational uses arises again and again, and these need to be reconciled.

Permitted sports activities in the Zentralfriedhof include running, Nordic walking and cycling. Running is a tranquil sport; experts consider running a form of active meditation. It is well suited to the cemetery, which has plenty of space. In the past, many people used to run in the cemetery, but now they are actively invited to use the wider running track along the kilometer-marked paths. The cemetery tries to be as open as possible to innovation. For instance, a few years ago, the burial regulations stated that “ball games are prohibited”, but this passage has since been deleted.

There is also an annual Biodiversity Day at the cemetery. Several projects are also run in the cemetery in cooperation with the City of Vienna and other organizations, e.g., “Biodiversität am Friedhof”. They work together with several educational institutions to make the cemetery’s wildlife as accessible as possible to the public and children [32] (Figure 6).

### 4.2. The Result of the Analysis of the Sample Area

Based on the interviews and literature sources, the habitat characteristics of the three cemeteries studied were summarized, verified and refined during the field visits (Table 1). All three cemeteries have exceptional fauna, flora and fauna. However, none of them is protected as a nature reserve, nor are there any individually protected values on their territory. The Nemzeti Sírkert and the entire Zentralfriedhof are protected as historical monuments. We have summarized the cemeteries’ activities related to green space, habitat conservation and enhancement. It is clear that the three cemeteries have similar characteristics in many respects, so the improvements in the Zentralfriedhof can serve as a model for other cemeteries in Budapest.

## 5. Discussion

We compared our results with those of other studies and examined how they can be interpreted in relation to previous studies and the sources we have used. We have endeavored to evaluate the results in the broadest possible context.

### Ecological Solutions

Based on international sources [4,6,38], we have broken down the environmental value of cemeteries into the following subcategories: spatial context, burial type, built elements, plant application and habitat provision (Table 2). The authors highlight the need for greater attention to be paid to ecological solutions in cemeteries, with particular emphasis on vegetation management, which cannot be overemphasized as an element of climate change mitigation. Based on these aspects, we have summarized the ecological solutions that should be used in the design, maintenance and development processes and examined their presence in the sample sites.

Spatial context: Cemeteries are not considered in planning and strategy documents as facilities that reinforce the green space system and complement ecological corridors. Today’s urban cemeteries are generally not intensively connected to their surroundings, existing as isolated objects. However, these areas are characterized by their exceptional natural value. They can serve as a link (ecological stepping stone or corridor) between other protected areas of higher natural value, allowing the migration of plant and animal species.

Burial method: The different burial methods have different spatial and maintenance requirements. In Hungarian and Austrian cemeteries, space for traditional burial still predominates, but other forms of burial are also gaining ground and opportunities. For example, cremation as a form of burial requires less space and can be used to reduce the size of the burial area within the cemetery. Scattering ashes is also a more ecological and space-saving form of burial than urns, and using a scattering cemetery can create a representative green space. In forest cemeteries, scattered or small grouped graves are placed under an existing forest stand. In the case of memorial forests, even grave markers are not used in the cemetery, where the deceased are buried in urns that decompose under individual trees.

Planting scheme: Architecturally designed cemeteries often have spectacular tree lines along the main roads, providing shade for road users and adding to the aesthetic appeal of the environment. Where different species are used, they also help orientation between similar plots. In the case of park cemeteries, planting ornamental trees and shrubs between the graves is typical of public parks. In woodland cemeteries and memorial woods, native species of associative trees predominate. Evergreen species are planted in cemeteries in many cultures because of their symbolism. However, a wide variety of herbaceous species also appear, especially on graves. The proportion of ornamental plants in the cemetery flora can be very high, so the potential for plant invasion in these habitat patches is a major risk. Most ornamental plants are not necessarily alien or invasive. Most of them are not able to influence native vegetation. Still, they may alter habitat structure, or some sensitive native species may not be able to compete with successful non-native plants. Non-native woody species, for example, often dominate the cemetery landscape for decades. For cemeteries to become a biodiversity base and enrich the urban environment, the prioritization of native species (both woody and herbaceous) should be an important consideration.

Cemeteries provide ecosystem services in the same way as traditional urban green spaces, and their ecosystem services are similar: air quality, regulation of local climate and water balance, reduction of urban heat island effects, wood fuel from tree care, habitat for pollinators, and provision of, for example, aesthetic and recreational values.

Based on our literature research, we have compiled an assessment table identifying the characteristics of cemeteries, their ecological importance and the ecosystem services they provide. The assessment was carried out in all three sample areas. As quantitative data were not uniformly available for all three sites, we used exploratory estimation to classify the data associated with each cemetery into three categories depending on the extent to which the value was specific in the case of each cemetery (Table 2).

Table 2 shows that a wide range of ecological solutions can be applied in cemeteries to enhance their habitat function. Habitat improvements also contribute to the enhancement of the ecosystem functions of cemeteries, primarily regulating and maintaining. Based on the interview responses, the sources and documents explored and the field visit, it can be concluded that most of the habitat improvements that have been implemented worldwide are already in place or planned to be implemented in the case of the Zentralfriedhof Vienna. In Budapest, their use is currently limited in the Nemzeti Sírkert, but the cemetery development concept has already proposed the introduction of a number of solutions that could help to enhance the habitat function of the cemetery in the future. In the case of the Új Köztemető, the conditions (size, amount of open space) for habitat development are in place, but no decision has yet been taken on their implementation.

From our assessment results (Table 3), it is clear that all three cemeteries need further improvements to strengthen their ecosystem services. The Zentralfriedhof scored highly in most categories, with the most potential for improvement in the built elements. For the two cemeteries in Budapest, their size and current use had a significant impact on the results: for both the greatest potential for improvement is also in the structural elements, while for educational functions, the Nemzeti Sírkert clearly shows the results of recent years’ sensitization efforts. In terms of designing, planting and maintaining vegetation, both sites in Budapest have outstanding potential for improvement.

The majority of the ecological solutions identified can be applied to any cemetery, and can be implemented to ensure habitat function. However, the increasingly popular “natural burial” has a number of ecological benefits in itself, which (based on international experience) are already present in the design of such plots. Clayden et al. (2018) [42] investigate how natural burial transforms the traditional cemetery towards a more habitat-rich and spatially complex landscape, and also explore how natural burial increases the burial capacity of urban cemeteries. The ecological benefits of natural burial have been shown to reduce the total area of mown grassland, the frequency of mowing, and the use of herbicides and to create more complex habitats, including new woodlands. This potentially increases carbon sequestration and reduces NO2 emissions. Less-managed grasslands have a greater capacity for wildlife. Within the cemetery, marginal areas that are not suitable for traditional graves with headstones can be used. The recreational value of the cemetery is increased, encouraging people to visit. It may also have the effect of making the public more accepting of less intensively managed, aesthetically untidy and less well-maintained green spaces. Finding space for “wilder”, “untidier” nature is also difficult because it can create a sense of neglect on the part of cemetery users [42].

## 6. Conclusions

Cemeteries integrate elements of the natural and cultural environment: burial sites/burial places, vegetation, built elements and transport systems. Because of their natural and built character, an integrated approach (spatial and landscape context) is required in planning, design and maintenance. Environmentally friendly solutions will contribute to the conservation and enhancement of the cemetery’s wildlife and its recreational potential. In addition to environmental benefits, ecological approaches also bring social and economic benefits, which are summarized in the following table (Table 4).

Cemeteries can contribute to the conservation of biodiversity in cities. However, little is known about the diversity and composition of the plant and animal species that live in them, and researchers mainly focus on the prominent, easily detectable animal and plant species in cemeteries [39]. Thus, research in this area focuses mainly on woody stem plants [40] and bird species [41]. This was also observed in our sample plots. As most cemeteries have both natural and cultural heritage values, a better understanding of the biodiversity composition, the relationship and the needs of each species could facilitate conservation approaches. There are many examples, such as the Ohlsdorf cemetery, where development plans take into account both cultural and natural values. In the course of development, the cemetery is divided into intensively and extensively maintained areas in order to properly preserve their cultural and natural values [46].

The role of dead trees in habitat conservation was highlighted. Dead, standing tree trunks are generally absent from cemeteries, even though they are particularly important habitats for hundreds of species: fungi and insects, which depend on the different stages of decomposition of trees. In addition, the associated cavities provide a habitat for bats, woodpeckers and other cavity-dwelling insects. Dead wood is also a limited resource in urban parks, although it is an important tool for biodiversity conservation. However, this appearance does not necessarily correspond to the aesthetic preferences of urban residents and visitors to cemeteries. The ecologically valuable elements of green spaces and cemeteries, which are at odds with cultural norms, often create a sense of “disorderliness”. While people like living, old trees for the many associated services they provide (e.g., aesthetics or shade), this does not necessarily apply to dead trees [2]. It can be argued that social needs and aesthetics do not necessarily match ecological benefits and biodiversity conservation, and this poses a new challenge for urban planners and green space designers.

From our previous study [1], we know that urban residents visit cemeteries for different reasons and that different characteristics of cemeteries, especially trees/vegetation and the level of maintenance, influence the time of visit. Rather than a “one-size-fits-all” strategy for the whole cemetery area, our study advocates an approach that addresses different needs by designing differently maintained parts of the cemetery. In this way, urban cemeteries provide an opportunity to maintain and nurture urban flora and fauna, which greatly supports biodiversity conservation.

Urban cemeteries provide important ecosystem services, but little is known about the specific factors that generate these benefits. This key topic is still relatively under-researched.

Climate change challenges [42]: We need to rethink how we manage urban green spaces to help create stronger and more resilient cities. We need to increase habitat diversity and reduce traditional maintenance by reducing the amount of land mown, the frequency of mowing and the use of herbicides and by creating more complex habitats, which could include new woodland. This has the potential to increase habitat diversity and carbon sequestration and contribute to enhancing ecosystem services.

In the future, grasslands could also become an unaffordable luxury in parks. Natural grasslands and meadows provide a more complex habitat than agricultural land or traditional cemetery lawns. Natural, less intensively maintained areas within urban cemeteries can play a role in public acceptance of less intensively managed, aesthetically less tidy and less well-maintained public parks to achieve greater ecosystem services.

The educational and cultural value/role of cemeteries would be enhanced if special plants were “labeled” and animal habitats were signposted. This would help to promote the discovery and knowledge of native and exotic plant and animal species.

Tools for cemetery values and habitat conservation, based on our research and experience, and following [47], can be summarized in the following points:Value assessment—identification;Raising public awareness about the values of the site;Dissemination of information;Diversification of burial customs and maintenance technologies;Avoiding chemical weed control;Reduction of invasive plants;Preference for native plants;Community involvement in planning and in maintenance.

The question is how to reconcile the different objectives, including the preservation of cultural heritage, the possibility of continuous use (visits, burials) and the conservation of biodiversity. The answer can be found in a planning method that has been used in international practice for many years in the case of many cultural and natural sites: conservation management plans (CMPs) might be a suitable tool to achieve these goals.

During our research, it has emerged that there is no complex database available on urban park cemeteries. A few cemeteries have basic tree inventories, and/or surveys on endemic plant and animal species, but there is no database on cemeteries as complex systems, where the aspects of heritage, conservation, tourism and ecology are managed together. We would like to use the results of our research to lay the foundations for working out what data need to be collected for complex site developments. Soon we plan to set up a pilot project involving biologists, botanists, heritage conservators and representatives of stakeholders for formulating complex development proposals that help the management and maintenance of cemeteries with regard to strengthening their ecological services.

## Figures and Tables

**Figure 1 plants-12-01269-f001:**
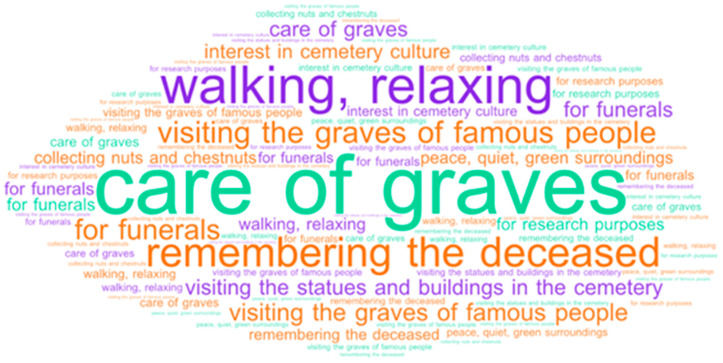
Word cloud: the diversity of purposes for visiting a cemetery, based on [1] (by authors).

**Figure 2 plants-12-01269-f002:**
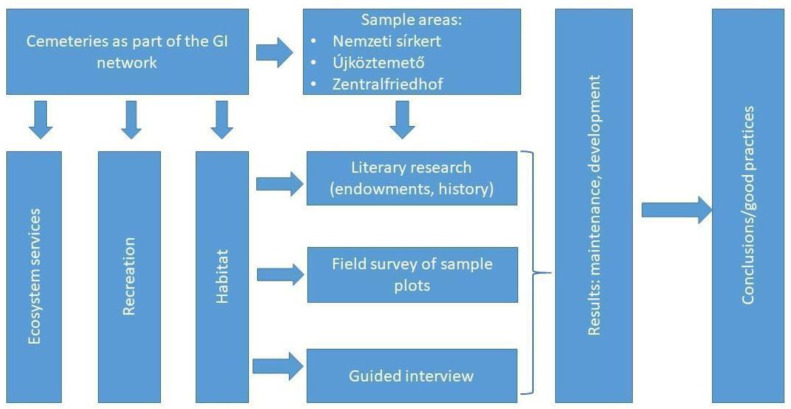
The structure of the study (by authors).

**Figure 6 plants-12-01269-f006:**
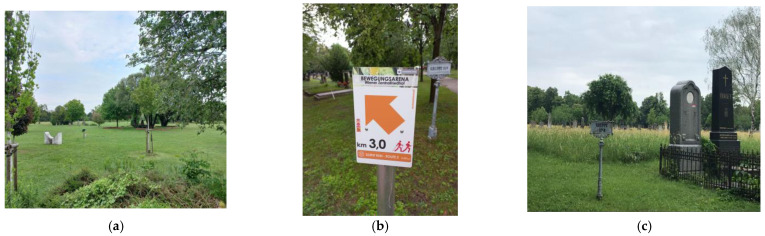
Zentralfriedhof Wien: (**a**) Park der Ruhe und Kraft/Park of peace and strength, (**b**) running route kilometer board, (**c**) bee pasture (authors’ photos, 2022).

**Table 1 plants-12-01269-t001:** Summary of the three sample area cemeteries (authors’ table).

	Nemzeti Sírkert (1847)	Új Köztemető (1886)	Zentralfriedhof (1874)
Size	56 hectares	207 hectares (largest cemetery in Hungary)	291 hectares (2nd largest in Europe)
Style	architectural cemetery	architectural cemetery	architectural cemetery
Management	National Heritage Institute	Budapest Funeral Institute	Friedhöfe Wien GmbH
Status	functioning and closed (mostly) parts	functioning (mostly) and closed parts	functioning, closed parts and reserve area
Location in the city	in the urban fabric (earlier at the time of creation, on the outskirts of the city)	at the outskirts of the city	at the outskirts of the city
Accessibility	public transport, vehicle access allowed, no separate parking	public transport, vehicle access allowed, no separate parking	public transport, vehicle access allowed, several parking zones in the cemetery
Delimitation/entrances	solid brick wall with several gates, but only the main entrance available; no representative reception area	solid brick wall with several gates, but only the main entrance available, representative reception area at the central gate	solid brick wall with several gates, representative reception area at the central gate
Special parts of the cemetery	artists’ plots, academic plots, heroes’ plots, labor movement plots, Soviet military plots, military plots	heroes’ plots (plots 298, 300 and 301), military plots, monastic plot, Mohammedan plot, urn pantheon	children’s graves, forest cemetery, Nature Garden, Garden of Serenity and Strength, military plots, religious plots
Green space elements	Significant vegetation: alleys, landscaped plots, but also overgrown plots	Significant vegetation: alleys, lot of overgrown plots	Significant vegetation: alleys, landscaped plots, overgrown plots and forest cemetery
Recreational areas	not specified	not specified	designated areas
Green surface (reserved)	ca. 67%	ca. 25%	ca. 75%
Green space intensity	ca. 83%	ca. 91%	no data
Green infrastructure connections	green area of the Jewish cemetery on Salgótarjáni road	green areas of the Jewish cemetery of Kozma Street and the Orthodox Jewish cemetery of Granatos Street; the Keresztúr forest	green area of the Protestant cemetery Simmering and new Jewish cemetery; part of the Vienna Green Belt
Animals	birds (110 resident bird species and 40 breeding bird species), insects, bats, squirrels, other rodents, hedgehogs, foxes	50-head deer herd, birds, insects, bats, squirrels, other rodents, hedgehogs, foxes, field rabbits	birds (e.g., ducks), squirrels, field hamsters, other rodents, roe deer, stags, bees, butterflies and other insects, amphibians, hedgehogs, rabbits
Habitat conservations status and potentials	bird watching and ringing, landscaped and overgrown plots	overgrown plots	landscaped and seminatural plots, forest cemetery
Habitat developments	bird holes	nature trail in the Keresztúr forest	beehives, Nature Garden (small pond and butterfly meadows, flower hedges, trees, deadwood pile, insect hotel), natural meadow for insects, bird holes, project for the protection of hedgehogs, ”Biodiversität am Friedhof” program, deadwood program
Buildings	cemetery office, museum; administration building; chapel; funeral parlor; mausoleums and arcades	cemetery office; visitor center at plot 301; funeral parlors and crematorium; bell tower; arcades; operational buildings	cemetery office; museum; funeral parlors; cemetery chapel(s); arcades; operational buildings; solar park
Works of art	mausoleums, tombs, gravestones, statues	tombs, gravestones, statues	mausoleums, tombs, gravestones, statues
Significance	yes, national pantheon	yes, Hungary’s largest cemetery	yes, Vienna’s most important cemetery
Touristic offers	thematic guided walks and mobile application	thematic guided walks	thematic guided walks on foot, horse-drawn carriage or e-bike; mobile application

**Table 2 plants-12-01269-t002:** Environmental (habitat) values of cemeteries (authors’ table).

Analytical Aspects and Criteria	Description	Significance by Means of Environmental/Ecological Benefits	Associated Ecosystem Service	Is It Present in the Cemeteries Surveyed?
SPATIAL INTEGRATION—INTEGRATION INTO GREEN INFRASTRUCTURE				
Location of cemeteries in the urban fabric	Distance and relationship to other green spaces, green corridors	Continuity in landscape/space. Provision of transport corridor and habitat for species	Regulator and management service	NS ⚘, ÚK ⚘⚘, Zf ⚘⚘⚘ ^1^
Accessibility	Accessible by public transport/by bike/on foot, parking spaces available	Increased interest, high visitor numbers, intensive care, cemetery development, higher maintenance costs	Regulator and management service. Cultural service	NS ⚘⚘, ÚK ⚘⚘, Zf ⚘⚘
Design of the cemetery area	Design, internal layout adaptation to natural features (e.g., topography, hydrography–watercourse)	Minimal impact on topography, reduction of earthworks. Preservation of native vegetation. Increased space for green areas in the design of the cemetery	Regulator and management service	NS ⚘⚘, ÚK ⚘⚘, Zf ⚘⚘⚘
BURIAL TYPE				
Traditional coffin burial	Covered or framed graves, high proportion of paved surface	In the case of framed graves, it is possible to plant species on the graves	Regulator and management service	NS ⚘⚘, ÚK ⚘, Zf ⚘⚘
Cremation—urn burial	Urn placement solutions: urn walls, courtyards, urn graves	Smaller burial space, possible increase in planted areas covered by vegetation	Regulator and management service	NS ⚘⚘, ÚK ⚘⚘⚘, Zf ⚘
Cremation—ashing	After cremation, scattering the ashes in a spreading parcel (or in water, e.g., in the Danube)	Grassland, near-natural, native herbaceous plants can be used for landscaping	Regulator and management service	NS ⚘⚘, ÚK ⚘⚘⚘, Zf ⚘
Gravestone/”headstone” funerals (both coffins and urns)	(standardized) Gravestones or memorial plaques flush with turf and grassland areas	Maximum beneficial use of rainfall, smaller burial area. Requires intensive maintenance, but can be mechanized. Low level of biodiversity.	Regulator and management service	NS ⚘⚘⚘, ÚK ⚘⚘⚘, Zf ⚘⚘⚘
Forest cemetery, park cemetery	The graves are either scattered in small groups in the landscaped area or are planned to be located under the forest that was already there	More green space, more opportunities for recreational and ecological functions	Regulator and management service	NS ⚘, ÚK ⚘, Zf ⚘⚘
Memorial forests, memorial gardens	Separate space dedicated to different solutions of burial, other than the traditional forms of it; e.g., urns placed around trees without grave signs	Higher plant coverage, use of decomposable urns, minimum presence of built elements and artificial materials	Regulator and management service	NS, ÚK, Zf ⚘⚘⚘
ARCHITECTURE, BUILT ELEMENTS				
Funeral parlor, cemetery chapel, administrative buildings	Ecological architecture, sustainable buildings	Well integrated into the environment, prioritizing the protection of natural resources, lower maintenance and heating costs and emissions, raising social awareness	Regulator and management service	NS ⚘, ÚK ⚘, Zf ⚘⚘
Supporting structures for creeping plants along the walls	Use of climbing plants on building facades (green walls)	Reduction of heating or cooling energy costs of buildings Pleasant view + habitat/shelter	Regulator and management service	NS, ÚK, Zf
Green roofs	Planting special species on roofs	Natural insulation, additional biologically active area	Regulator and management service	NS, ÚK, Zf
Lighting	Solar, LED lighting	Energy reduction, lower maintenance costs, lower light pollution	Regulator and management service	NS, ÚK, Zf ⚘⚘⚘
Covering of pavements and transport areas	Use of permeable technologies for paving surfaces	No run-off water, most precipitation quantity is used locally	Regulator and management service	NS ⚘, ÚK ⚘, Zf ⚘
Hydrological features	Reservoirs, rain gardens	Reduced water run-off and precipitation loss. More pleasant environment, microclimate valorization (humidification, temperature reduction). Habitat creation	Regulator and management service	NS, ÚK, Zf ⚘
Fencing, enclosure	Use of natural materials, providing passage for animals	Increase in biodiversity, passage for animals	Regulator and management service	NS ⚘, ÚK ⚘, Zf ⚘
PLANT USE AND APPLICATION				
Lawn surfaces	Homogeneous plant application, intensive maintenance (irrigation, mowing)	Low species richness, but biologically active surface	Regulator and management service	NS ⚘⚘⚘, ÚK ⚘⚘, Zf ⚘⚘⚘
Perennials	Flowering meadows, native plants	Lower maintenance costs, reduced labor requirements	Regulator and management service	NS ⚘, ÚK ⚘, Zf ⚘⚘
Shrubs and bushes	Hedges, e.g., around plots, along fences—functional and aesthetic role	Prevent monocultures, reduce erosion, increase biodiversity	Regulator and management service	NS ⚘, ÚK ⚘, Zf ⚘⚘
Trees	Alleys, clumps, solitary trees, woody areas	Improved air quality, wind strength and wind erosion reduction, shading–temperature reduction	Regulator and management service	NS ⚘⚘⚘, ÚK ⚘⚘⚘, Zf ⚘⚘⚘
Creeping plants	Creeping species covering buildings, columns, supports and structures	Increased biodiversity and shading–temperature reduction	Regulator and management service	NS ⚘, ÚK ⚘, Zf ⚘⚘
Plant diversity	Complex vegetation coverage, use of native species	Soil structure and permeability improved by roots, higher water supply, increase in air moisture, increase in biodiversity, habitat creation	Regulator and management service	NS ⚘, ÚK ⚘, Zf ⚘⚘⚘
HABITAT PROVISION				
Extensively used and reserved areas	Less frequent mowing, leaving mowed grass and meadow in peace, leaving cut-off branches in situ	Providing habitat for a range of animal and plant species. Increased biodiversity	Regulator and management service. Supply service	NS ⚘⚘, ÚK ⚘⚘, Zf ⚘⚘
Artificial habitats	Installation of an insect hotel, bird house, bat house, etc.	Creating better living conditions for certain animal species. Increased biodiversity	Regulator and management service. Supply service	NS ⚘⚘, ÚK, Zf ⚘⚘⚘
Bee pastures	Less mowing, use of flowering plants to provide nutrients for pollinators	Not only a feeding ground, but also a hiding and breeding place	Regulator and management service. Supply service	NS ⚘, ÚK ⚘, Zf ⚘⚘
Water surfaces	Can be used to collect and store rainwater. In addition, it provides aesthetic and ecological benefits	Habitat, feeding and breeding ground, special microclimate, aesthetic pleasure	Regulator and management service. Supply service	NS, ÚK, Zf ⚘⚘
Beehives	Can be hosted in less visited areas, subject to local regulations	It has a positive impact on plant development and biodiversity, and also generates income through diverse products	Supply service	NS, ÚK, Zf ⚘⚘⚘
In situ decayed trees	Retention of decomposing, dying trees	Improved habitat creation, but might be less aesthetic	Regulator and management service	NS ⚘, ÚK ⚘, Zf ⚘⚘⚘
AWARENESS RAISING AND EDUCATION				
Information boards and panels	Information and knowledge sharing in case of interesting plants and habitats	To familiarize visitors with the species, the importance of the habitats and the associated conservation practices	Cultural service	NS ⚘⚘, ÚK ⚘, Zf ⚘⚘⚘
QR code bars, dedicated applications	Provided information on specific species, habitats and developments using smart devices	Tourist attraction, awareness raising, increasing number of visitors	Cultural service	NS ⚘⚘⚘, ÚK ⚘, Zf ⚘⚘⚘

^1^ NS: Nemzeti Sírkert, ÚK: Új Köztemető, Zf: Zentralfriedhof; ⚘: present but not dominant in habitat (less than 40%); ⚘⚘: present but in need of development/enhancement (40–70%); ⚘⚘⚘: dominant element in the cemetery (more than 70%).

**Table 3 plants-12-01269-t003:** Environmental (habitat) values of cemeteries—summary (authors’ table).

	1. Spatial Integration—Integration into Green Infrastructure (Max. 9)		2. Burial Type (Max. 18)		3. Architecture, Built Elements (Max. 21)		4. Plant Use and Application (Max. 18)		5. Habitat Provision (Max. 18)		6. Awareness Raising (Max. 6)	
	⚘	%	⚘	%	⚘	%	⚘	%	⚘	%	⚘	%
NS—Nemzeti Sírkert	5	55	10	55	3	14	10	55	6	33	5	83
ÚK—Új Köztemető	6	66	11	61	3	14	9	50	4	22	2	33
Zf—Zentralfriedhof	8	88	12	66	8	38	15	83	15	83	6	100

⚘ indicates the extent of habitat value.

**Table 4 plants-12-01269-t004:** Environmental, social and economic benefits of green solutions (authors’ work).

Environmental	Social	Economic
appropriate maintenance, “no intensive maintenance”: less frequent mowing less tampering leaving branches that may have been cut artificial habitats, e.g., insect hotel “unpaved” roads near-natural improvements: unpaved footpaths, bee-keeping areas, other biotopes lighting	less disruption acceptance of nature-based maintenance interest in wildlife education community building	guided nature walks bees, apiary honey production and sale collection and sale of medicinal plants lower maintenance costs, in the case of “unmanaged” areas, even solar energy recovery

## Data Availability

Not applicable.

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
