# Peer review of "The Role of Urban Cemeteries in Ecosystem Services and Habitat Protection"

_plants, 2023, doi:10.3390/plants12061269_

Round 1

Reviewer 1 Report

- To the introduction or literature review, I would recommend adding more literature sources on cemeteries as urban green spaces.

-The title suggests that the main research topic is cemeteries as shelter habitats, but the research is broader - as written in the aims and research question (lines 124-136). I would recommend either revising the title or accentuating the shelter habitat aspect in the text.

Author Response

Dear Reviewer, Thank you for your thorough work and the time you took to prepare this review. We thank you for your many positive comments, your praise and overall positive assessment of the study and your suggestions for further reflection. We agree with many of your points. The constructive critical comments have contributed to the higher quality of our article. Below, we respond to the comments, suggestions and critiques made in the review.

Our overall changes to the article: The study has undergone a major structural revision. It has been given a stronger focus and deeper literary support. We have revised the article to make the methodology clearer and more complete. We have also revised and completed the results section and modified the title as well.

Reviewer 2 Report

The authors present an interesting in depth historical and qualitative description of the study areas and a detailed review of the literature concerning the sites and environmental services of cemeteries. Their merit is to identify the importance of these types of open spaces in the context of the green infrastructure.  However the lack of innovative research approaches and methods and the lack of any quantitative results and analysis, makes the study of limited value in an international context and in the context of this journal. In my opinion the authors could find a more suited journal then Plants for publication, or they should extensively revise the paper proposing a quantitative analysis of their main findings.

Author Response

Dear Reviewer, Thank you for your thorough work and the time you took to prepare this review. We thank you for your many positive comments, your praise and overall positive assessment of the study and your suggestions for further reflection. In our opinion, our article fits well in the special issue "Ornamental Plants and Urban Gardening" of the Plants journal, as it deals with a specific green infrastructure element with a high potential for cities: the urban cemeteries. For cemeteries in general, and for Hungarian cemeteries in particular, it is a problem to obtain quantitative data, as cemeteries collect only limited data on their vegetation and fauna. With our research we wanted to lay the foundations for further studies and data collection that would take into account the complex role of cemeteries, which in addition to their role in memorial, cultural history and tourism also places great emphasis on their ecological role. We agree with many of your points. The constructive critical comments have contributed to the higher quality of our article. Below, we respond to the comments, suggestions and critiques made in the review.

Our overall changes to the article: The study has undergone a major structural revision. It has been given a stronger focus and deeper literary support. We have revised the article to make the methodology clearer and more complete. We have also revised and completed the results section and modified the title as well.

Round 2

Reviewer 2 Report

The authosr have more clearly stated the purpose of the study and have improved the structure of the paper.  I still think the paper would have been more appropriate for a journal specifically dealing with green infrastructure and urban sustainability, nonetheless the quality of the paper is good so I have no objections to its publication in the work in its present form